# Oxidative Stress in Long-Term Exposure to Multi-Walled Carbon Nanotubes in Male Rats

**DOI:** 10.3390/antiox12020464

**Published:** 2023-02-12

**Authors:** Ewa Florek, Marta Witkowska, Marta Szukalska, Magdalena Richter, Tomasz Trzeciak, Izabela Miechowicz, Andrzej Marszałek, Wojciech Piekoszewski, Zuzanna Wyrwa, Michael Giersig

**Affiliations:** 1Laboratory of Environmental Research, Department of Toxicology, Poznan University of Medical Sciences, 60-631 Poznan, Poland; 2Faculty of Chemistry, Adam Mickiewicz University, 61-614 Poznan, Poland; 3Centre for Advanced Technologies, Adam Mickiewicz University, 61-614 Poznan, Poland; 4Department of Orthopedics and Traumatology, Poznan University of Medical Sciences, 61-545 Poznan, Poland; 5Department of Computer Science and Statistics, Poznan University of Medical Sciences, 60-806 Poznan, Poland; 6Oncologic Pathology and Prophylaxis, Greater Poland Cancer Centre, Poznan University of Medical Sciences, 61-866 Poznan, Poland; 7Department of Analytical Chemistry, Faculty of Chemistry, Jagiellonian University, 30-387 Krakow, Poland; 8Department of Theory of Continuous Media and Nanostructures, Institute of Fundamental Technological Research, Polish Academy of Sciences, 02-106 Warsaw, Poland

**Keywords:** multi-walled carbon nanotubes, oxidative stress parameters, rats, long-term toxicity

## Abstract

Multi-walled carbon nanotubes (MWCNTs) serve as nanoparticles due to their size, and for that reason, when in contact with the biological system, they can have toxic effects. One of the main mechanisms responsible for nanotoxicity is oxidative stress resulting from the production of intracellular reactive oxygen species (ROS). Therefore, oxidative stress biomarkers are important tools for assessing MWCNTs toxicity. The aim of this study was to evaluate the oxidative stress of multi-walled carbon nanotubes in male rats. Our animal model studies of MWCNTs (diameter ~15–30 nm, length ~15–20 μm) include measurement of oxidative stress parameters in the body fluid and tissues of animals after long-term exposure. Rattus Norvegicus/Wistar male rats were administrated a single injection to the knee joint at three concentrations: 0.03 mg/mL, 0.25 mg/mL, and 0.5 mg/mL. The rats were euthanized 12 and 18 months post-exposure by drawing blood from the heart, and their liver and kidney tissues were removed. To evaluate toxicity, the enzymatic activity of total protein (TP), reduced glutathione (GSH), glutathione S–transferase (GST), thiobarbituric acid reactive substances (TBARS), Trolox equivalent antioxidant capacity (TEAC), nitric oxide (NO), and catalase (CAT) was measured and histopathological examination was conducted. Results in rat livers showed that TEAC level was decreased in rats receiving nanotubes at higher concentrations. Results in kidneys report that the level of NO showed higher concentration after long exposure, and results in animal serums showed lower levels of GSH in rats exposed to nanotubes at higher concentrations. The 18-month exposure also resulted in a statistically significant increase in GST activity in the group of rats exposed to nanotubes at higher concentrations compared to animals receiving MWCNTs at lower concentrations and compared to the control group. Therefore, an analysis of oxidative stress parameters can be a key indicator of the toxic potential of multi-walled carbon nanotubes.

## 1. Introduction

Nanomedicine is currently used for the development of various medical products, and it has revolutionized the diagnosis and treatment of numerous diseases. Particle sizes in nanotechnology are similar in size to receptors, antibodies, or biomolecules. Their ability to modify and their high surface-to-volume ratio makes nanoparticles, including nanotubes, an excellent tool for use in medicine [1]. One direction of research in the field of new surgical techniques and biomaterials is the creation of biocompatible scaffolds based on natural and synthetic chemical compounds to repair different types of tissue. Toxicological studies of biomaterials are also required, and they are currently being carried out by our team, among others. Nanoparticles offer many advantages in the medical field, but the properties that make them attractive for application can also affect their toxicological profile in biological systems, making their size, shape, and chemical composition worth considering in nanoparticle production. There are also concerns about cellular network interactions, the endocytic pathway, and the absorption process, which can also induce cytotoxicity, leading to the disruption of cellular homeostasis [2]. One of the key factors in the interaction with biological systems of nanoparticles is their size, which is strongly associated with toxic effects. Smaller nanoparticles have a greater surface area per unit of mass and are, therefore, able to absorb a large number of chemical molecules. MWCNTs’ diameters range from 5 to 20 nm, although polyhedral diameters can exceed 100 nm, mainly depending on the number of layers of nanotube walls and functional groups attached to them [3]. This results in increased reactivity in the cellular environment and, thus, greater toxicological effects [4]. Toxicity is also influenced by the surface area of the nanoparticles due to its relationship with absorption efficiency [5]. Nanotoxicity, thus, seeks to establish the level or extent to which these properties may pose risks to the environment or to the life of organisms. The reason to start designing nano-drugs is to reduce the toxicity of the drug and increase its bioavailability and biocompatibility. On the other hand, it must be reckoned that their specific properties may pose a risk to patients. Nanoparticles show toxicity through various mechanisms and can lead to allergy, fibrosis, and organ failure. Their influence can embrace neurotoxicity, hepatotoxicity, nephrotoxicity, or pulmonary toxicity [6]. The toxicity of nanoparticles depends on a number of factors, such as purity, type of synthesis, coating, shape, concentration, or biological system tested [7]. Carbon nanotubes (CNT) are allotropes of carbon and can be viewed as a two-dimensional hexagonal lattice of carbon atoms (graphene) rolled seamlessly into a cylinder. Depending on morphology, several types of CNTs can be distinguished, e.g., single-walled carbon nanotubes, multi-walled carbon nanotubes, and bamboo-type carbon nanotubes. Each type may vary in aspect ratio, symmetry, chirality, and surface chemistry and, therefore, display various physicochemical properties, directly affecting toxicity. CNTs have been successfully used in several tissue engineering applications, including 3D bioprinting [7,8,9,10,11,12,13].

The internal toxicity of CNTs depends on the degree of surface functionalization and the toxicity of the functional groups. Another important factor in CNTs’ toxicity is their bioavailability. Metabolism, degradation, dissolution, clearance, and bioaccumulation require attention and research to understand the limitations of nanomaterials as pharmaceuticals [14,15,16,17,18,19,20,21,22,23,24,25,26,27]. One of the challenges in using carbon nanotubes has been the issue of biocompatibility. Once CNTs enter the body, they interact with body fluids and organs. The main routes of penetration of carbon nanotubes into the human body are the mouth, nose, or skin, targeting the gastrointestinal tract, respiratory system, or causing skin erosion, respectively [28]. According to research, CNTs exhibit certain levels of toxicity depending on the organ [29]. The biodistribution of CNTs in the body can lead to different toxicities depending on their concentration, components, structure, size, and functionalization [29]. Importantly, the investigation of CNTs toxicity is also hindered by the interference of the CNTs with luminescence-based assays, widely used for the determination of various cytotoxic effects [30].

The liver is one of the most important organs in the body and plays a key role in the metabolism of xenobiotics. It is, therefore, necessary to study and analyze the toxic effects of CNTs on the liver. Liver sinusoidal and Kupffer cells, as major structures for metabolism and detoxification, are susceptible to toxicity and nanoparticle deposition [31]. The liver accumulates 30–99% of nanoparticles from the bloodstream, which leads to increased hepatotoxicity. Kupffer cells, found in the liver, also play an important role in the immune system thanks to their phagocytic capacity, which allows them to capture and eliminate parasites or bacteria [32]. According to studies by Shedova et al. and Ahmadi et al., cellular detoxification pathways, such as apoptosis and antioxidants, are activated in response to nanomaterial administration, reducing cytotoxicity [33,34]. Repeated, short-term and intraperitoneal administration of purified carboxylated MWCNTs (diameter 15–30 nm, lengths 15–20 mm) has been shown to induce oxidative liver damage in pubertal rats through chemical and biological interactions, disruption of antioxidant defenses, increased cyclooxygenase-2 (COX-2) and nitric oxide synthase (iNOS) levels, and production of pro-inflammatory cytokines [35,36,37]. Studies suggest that organs that are involved in the excretion of toxic substances are more likely to accumulate CNTs [38]. The kidneys play a key role in the excretion of toxic substances and their metabolites from the body, so they are more likely to accumulate CNTs, which can lead to nephrotoxicity [38]. MWCNTs sized from 60 to 80 nm were shown to cause significantly greater nephrotoxicity compared to nanotubes sized from 90 to 150 nm [39]. The effects of different types of MWCNTs administered intravenously to healthy mice on the histology of various tissues, including the kidney, were also studied. The results revealed that a higher degree of ammonium (NH3) modification on the surface of the nanotubes resulted in lower accumulation in tissues. In addition, histological analysis of glomeruli 24 h after administration showed no change in glomerular physiology, and no accumulation was observed in all types of MWCNTs (diameter 20–30 nm, length 0.5–2 μm) tested, including primary, diethylentriaminepentaacetic (DTPA), and ammonium functionalized [40].

Oxidative stress was initially defined by Sies as a severe imbalance between oxidation and antioxidants [41]. An imbalance of reactive oxygen species and antioxidants in favor of prooxidants leads to potential damage. It is involved in cell signaling and regulation, but in excess, it can cause oxidative damage to cells [42,43,44]. Nanoparticles often cause cellular oxidative stress through the dysfunction of organelles such as mitochondria, peroxisomes, lysosomes, and the Golgi apparatus. This leads to the overproduction of ROS through the dysfunction of these organelles. Oxidative stress biomarkers are, therefore, important tools for assessing nanoparticle toxicity [45]. Oxidative stress induced by nanoparticles can be divided into two types based on their mechanism. One is direct oxidative stress, referred to as primary oxidative stress. It involves the direct induction of oxidative stress by ROS generated on the surface of nanoparticles. The other mechanism is indirect oxidative stress, otherwise known as secondary oxidative stress. It involves the generation of ROS due to mitochondrial dysfunction upon exposure to nanoparticles. In the mitochondrion, O_2_, H_2_O_2_, and hydroxyl radicals are constantly generated during the electron transport chain. Under physiological conditions, ROS are continuously removed by an oxidant system consisting of superoxide dismutase, catalase, and glutathione [46]. The mechanism by which nanomaterials induce oxidative stress in the living organism is a complex process, and both primary and secondary causes are usually associated with its induction [2,39,43,47,48]. Carbon nanotubes have often been studied to establish a pathway for them to enter cells through the double lipid layer of the cell wall. According to some studies, one of the cellular mechanisms of CNT uptake is endocytosis [49]. Penetration of the nanotubes through the lipid membrane causes oxidative stress, which can lead to an inflammatory response and also cause cytotoxicity. CNTs behave like a foreign body towards the cell, which causes chemical substances to be secreted and released to eliminate them from the cell [50]. Carbon nanotubes are not easily removed from the body, which entails a higher risk of accumulation in organs. According to some studies, organs such as the spleen, kidneys, and lungs are easy targets for free radical-induced oxidative stress [51]. Another mechanism responsible for the generation of toxicity is the formation of reactive oxygen species. Increased levels of ROS lead to detrimental effects on cells, such as apoptosis, damage to genetic material, oxidation of amino acids, and inactivation of enzymes. The increased inflammatory response following CNT exposure is also associated with the generation of toxicity within the body. The main reason for this response is so-called ‘frustrated phagocytosis’, in which macrophages are unable to engulf carbon nanotubes [52,53,54]. Some researchers even suggest that SWCNTs cause more apoptosis than MWCNTs [55,56]. One of the most popular materials used in scaffolds for tissue engineering is, indeed, carbon nanotubes. They have a high aspect ratio and a very wide range of possible dimensions, which makes them attractive for the fabrication of complex nanoarchitectures. Recent research is focused on the design and characterization of potential biomaterials for articular cartilage repair. Our research was conducted with this in mind for the future, so MWNCTs were administered to the knee joint to observe possible changes in the joint. Our animal model studies of multi-walled carbon nanotubes include measurement of oxidative stress parameters, including total protein (TP), reduced glutathione (GSH), glutathione S–transferase (GST), thiobarbituric acid reactive substances (TBARS), Trolox equivalent antioxidant capacity (TEAC), nitric oxide (NO), and catalase (CAT) in the body fluids, kidneys, and livers of animals as well as histopathological examination. Thus, the aim was to evaluate toxicity by assessing oxidative stress in this organ of male rats during long-term exposure using well-known oxidative stress biomarkers. Therefore, in this study, we compared toxicity in animals receiving MWCNTs in different concentrations with the control group during 12- and 18-month exposures.

## 2. Materials and Methods

MWCNTs with a diameter of 15–30 nm, length of 15–20 μm, and purity of up to 95% produced by chemical vapor deposition (CVD) were supplied by Nanolab Inc., Boston, MA, USA. The MWCNTs were functionalized and characterized as described in our previous work [30]. Briefly, MWCNTs produced using chemical vapor deposition (CVD) were treated with a mixture of concentrated sulfuric (H_2_SO_4_) and nitric (HNO3) acids in a ratio of 3:1 in an ultrasonic bath at 70 °C. After the reaction, the solution was neutralized with 3 M sodium hydroxide (NaOH). Oxidized carbon nanotubes were washed with a series of centrifugation in Milli-Q water. The resulting carbon nanotube solution was dried under vacuum and resuspended in a sterile phosphate buffer (PBS). Thermogravimetry was used to calculate the concentration of the oxidized MWCNTs. The physical and chemical parameters of MWCNTs were determined using scanning electron microscopy (SEM) and energy-dispersive X-ray spectroscopy (EDS) (Quanta 250 FEG, FEI Company, Hillsboro, OR, USA), and Fourier-transform infrared spectroscopy (FTIR) (Jasco 4700A—Jasco Corporation, Tokyo, Japan) (Figure 1).

### 2.1. Ethics Committee Approval

The study design was approved by the Ethics Committee for Animal Experiments Affairs in Poznań, Poland (Approval No. 9/2018; 61/2018; 36/2019). All procedures concerning the handling and use of laboratory animals were performed in accordance with European Union (UE) regulations under Directive 2010/63/EU on the protection of animals used for scientific purposes. Experiments were carried out in accordance with the so-called 3Rs principle (Replacement, Reduction, Refinement) to protect animals. In order to obtain consistent data, the study was based on the required minimum number of animals and observation time. To improve the rigor and reproducibility of animal research, all data will be collected according to ARRIVE 2.0 guidelines. In vivo experiments were carried out in the Animal House of the Wielkopolska Center for Advanced Technologies of the University of Adam Mickiewicz in Poznań and in the Animal House of the Department of Toxicology and Laboratory of Experimental Animals of the University Apparatus Center of the Poznan University of Medical Sciences in Poznań Poland. The above-mentioned centers are units listed by the Ministry of Education and Science. Contractors have individual permits for planning and performing experiments, as well as the killing of animals.

### 2.2. Animals and Experimental Treatments

The experiment was created in accordance with the guidelines of the Animal Protection Act used for scientific purposes and education (Journal of Laws 2015, of 26 February 2015, item 266), which was created on the basis of the EU Directive 2010/63.

The experiment was carried out on male Rattus Norvegicus/Wistar rats, outbred herd, 4 weeks old, with an average body weight of 75.0 ± 5 g. For the purposes of the experiment, animals of one sex (male) were used due to the minimization of the number of animals used in the experiment, in accordance with the 3R principle (Replacement, Reduction, Refinement), which aims to protect animals. We have limited the number of animals that need to be tested to a minimum to obtain reliable results. The latest statistical methods for elaborating the results for the minimum number of samples were used. The procedure and the activities planned in it have been developed in such a way as to minimize the suffering of the animals in the experiments as much as possible.

Among other things, due to the minimization of the number of animals used for the experiment, rats of one sex were selected. The proposed size of the study groups is the optimal number to maximize the scientific integrity of the generated data while using the minimum number of animals necessary for statistical calculations of test results and drawing appropriate conclusions from the conducted experiment. The animals were properly kept in polypropylene cages (*n* = 2 rat/cage) with autoclaved pine sawdust litter under controlled environmental conditions (12 h light/ 12 h dark: 6 am–6 pm; temperature: 22 ± 2 °C; air humidity: 50–60%). The animals were allowed to acclimatize for two weeks before the beginning of the experiment (Laboratory of Experimental Animals) with ad libitum access to water and wholesome feed. The water was sterilized before being given to the animals. The animals were fed Labofeed B Plant (“Morawski” Feed Production Plant—the dietary formula was created based on the recommendations of the National Research Council in the field of Nutrient Requirements of Laboratory Animals). After two weeks of acclimatization, the rats were randomly divided into eight groups of six animals each. Each animal was administered a single injection to the knee joint with a solution of multi-walled carbon nanotubes in a volume of 25 µL in a vehicle-buffered saline (PBS, Merck). Animals in each group were given 3 respective concentrations: 0.03 mg/mL, 0.25 mg/mL, and 0.5 mg/mL (6 animals in each group). A reference control group was administered 25 µL vehicle-buffered saline (PBS) (Figure 2). Then, 12 and 18 months after the administration of polyhedral carbon nanotubes to the knee joint, the animals a 1:1 (*v*/*v*) combination of ketamine (90 mg/kg; Kepro, Netherlands) and xylazine (10 mg/kg; Kela, Belgium) were injected intraperitoneally to induce anesthesia. A total of 10 min after injection when deep anesthesia was accomplished, animals were euthanized by drawing blood from the heart, and their liver (right lobe-histopathological analysis; left lobe-determination of oxidative stress parameters) and kidney (right kidney-histopathological analysis; left kidney-determination of oxidative stress parameters) tissues were removed. The fragments of each tissue were placed in 10% formalin solution, neutral buffered (Sigma Aldrich, St. Louis, MO, USA) for histopathological examination, and kept at room temperature. Additionally, to measure the enzymatic activity of TP, GSH, GST, TBARS, TEAC, NO, and CAT, each rat tissue was placed in PBS and kept in the freezer at −20 °C. During animal experiments, all ethical issues of working with animals were observed according to the guidelines of the Ethics Committee for Animal Experiments Affairs in Poznań, Poland.

### 2.3. Preparation of Blood and Tissue Samples

Biochemical determinations were performed in the Laboratory of Environmental Research at the Department of Toxicology at the Poznan University of Medical Sciences.

Blood collected in test tubes without anticoagulant (serum collection tubes, sterile, closed blood collection system, 4.5 mL, S-Monovette^®^, SARSTEDT) was left for 30 min in an upright position. Then, after 10 min of centrifugation at 3000 rpm, the clot was removed, and the serum was transferred to a new, sterile tube, secured, and stored at −80 °C until the analysis was performed.

A total of 1 g of liver tissue/0.8 g of kidney tissue were weighed. The biological material was divided into smaller parts and placed in 50 mL Falcon tubes. Then, 4 mL of PBS buffer diluted 1: 9 with saline was added to the test tube. Tissues were minced with a homogenizer (24,000 rpm) and transferred to 15 mL Falcon tubes. Then, the biological material was centrifuged for 10 min at 4 °C (4200 rpm). The obtained supernatant was transferred to 2 mL Eppendorf tubes and centrifuged again for 10 min at 4 °C (6000 rpm). The obtained supernatant was pipetted into Eppendorf tubes with a capacity of 2 mL. The biological material prepared in this way was stored in a freezer at −80 °C.

### 2.4. Determination of Oxidative Stress Markers and Biochemical Parameters

All chemicals used for biochemical determinations were of analytical reagent grade (Merck/Sigma Aldrich). Relevant markers of oxidative stress and biochemical parameters—TP, TEAC, NO, TBARS, GSH, GST were determined in serum (100 µL), liver (100 µL), and kidney (100 µL), and CAT was determined in the liver (100 µL), and kidney (100 µL), using spectrophotometric methods. Total protein (TP) concentration was determined using Lowry’s method—a combination of a biuret test and Folin-Ciocalteu reaction [57]. The Trolox equivalent antioxidant capacity (TEAC) of substances present in the solutions was measured based on the measurement of stable radical cation reduction capacity (ABTS)^•+^ [58]. By measuring the concentration of stable degradation products, nitrates (V) and nitrates (III) (nitrites) in an aqueous solution, NO concentrations were determined [59]. Thiobarbituric acid reactive substances (TBARS) measurement was used for monitoring lipid peroxidation [60]. Quantitative determination of reduced glutathione (GSH) was performed using modified Ellman’s method with 5,5′-dithiobis (2-nitrobenzoic acid) (DTNB, Ellman’s reagent) [61]. Glutathione S-transferase (GST) enzymatic activity was assessed based on the coupling reaction of thiol groups of L-glutathione with 1- chloro-2,4-dinitrobenzene (CDBN) (Bartosz, 2013). The activity of catalase (CAT) was determined based on the reaction of H_2_O_2_ degradation. The unit of CAT activity is the enzyme amount that degrades 1 µM H_2_O_2_ solution within 1 min, which corresponds to absorbance reduction by 0.036 U/min (volume: 1 mL, optical path length: 1 cm) [62]. 

### 2.5. Histological Analyses

Tissues were fixed in 10% buffered formalin for 24 h. Representative tissue sections were placed in histopathological cassettes and processed in a tissue processor. The tissue was dehydrated gradually through a series of ethyl alcohol solutions with increasing concentrations (80–99.8%), subsequently cleared in xylene, and embedded in paraffin. The paraffin-embedded tissue was cut into 5 μm sections using a rotary microtome (Accu-Cut^®^ SMRTM200, Sakura), placed on slides, and stained with HE and Masson’s trichrome using routine protocols.

### 2.6. Statistical Analysis

The calculations were made using Statistica 13 by TIBCO and PQStat by PQStat Software. The level of significance was α = 0.05. The result was considered statistically significant when *p* < α. The normality of the distribution of variables was tested with the Shapiro–Wilk test. To compare the variables between the 2 groups, in the case of compliance with the normal distribution and equal variances, the unpaired t-test was calculated; in the case of no variance equality—the Cochran–Cox test, and in the case of non-compliance with the normal distribution—the Mann–Whitney test. In order to compare the parameters between a larger number of groups, in the case of compliance with the normal distribution and equality of variance, the one-way ANOVA with Tukey’s multiple comparisons test was calculated. In the remaining cases, the Kruskal–Wallis test with Dunn’s multiple comparisons test was calculated. In order to determine whether the parameter concentrations in individual organs differ, the repeated measures ANOVA test with Tukey’s multiple comparison test or Friedman’s test with Dunn’s multiple comparison test was calculated. For the catalase, paired *t*-test or Wilcoxon’s test was calculated.

## 3. Results

### 3.1. Body Weights, Clinical Signs, and Food Consumptions

The treatment of knee-joint injected therapy with MWCNTs in different doses at two times of exposure caused no relative body weight change. Body weights were evaluated at the time of purchase and once a month during exposure. The animals were examined daily on weekdays for any evidence of exposure-related effects. Clinical signs and symptoms were not observed in the treated rats, and no significant change in food consumption was noticed during the exposure period.

### 3.2. Histological Analyses

#### 3.2.1. Kidneys

In the kidneys of male rats from the groups receiving MWCNTs at the concentration of 0.03 mg/mL, 0.25 mg/mL, and 0.5 g/mL, tubular dilatation was found in all age groups (12 and 18 months). Similar changes occurred in some of the control animals (Figure 3).

#### 3.2.2. Liver

Some of the animals showed parenchymal eclipse or focal bile duct proliferation—such changes occurred sporadically in the group of animals exposed to MWCNTs and control animals. The regularity of the changes could not be observed (Figure 3).

### 3.3. Concentrations of Oxidative Stress Markers and Biochemical Parameters

#### 3.3.1. Total Protein (TP)

##### Serum

The serum total protein (TP) levels in rats that were administered MWCNTs at a concentration of 0.5 mg/mL (94.36 ± 9.55 mg/mL) after 12-month exposure was significantly different from that of the control group (70.50 ± 17.38 mg/mL) and was 33.84% higher (Table 1).

The mean TP levels in animals given MWCNTs at a concentration of 0.25 mg/mL and euthanized after 18 months were significantly different from animals given MWCNTs at a concentration of 0.03 mg/mL and the control group. The TP levels in this group (49.32 ± 18.57 mg/mL) were 47.62% lower than the TP levels in animals given MWCNT at a concentration of 0.03 mg/mL (94.16 ± 19.52 mg/mL) and 44.16% lower than the serum TP levels in the control group animals (88.32 ± 28.33 mg/mL) (Table 1). There was a statistically significant increase of 45.72% in the serum total protein levels in the animals given carbon nanotubes at a concentration of 0.5 mg/mL (90.87 ± 18.44 mg/mL) compared to the animals given MWCNT at a concentration of 0.25 mg/mL (49.32 ± 18.57 mg/mL) (Table 1).

The research showed statistically significant differences in serum TP levels in rats receiving MWCNTs at a concentration of 0.25 mg/mL and euthanized after 18 months of exposure (49.32 ± 18.57 mg/mL) compared to rats receiving carbon nanotubes at the concentration of 0.25 mg/mL and exposed for 12 months (86.68 ± 26.45 mg/mL). TP levels in this group were 43.1% lower (Table 1).

##### Liver

During 18-month exposure, the research showed statistically significant differences in livers TP levels in rats receiving MWCNTs at a concentration of 0.5 mg/mL (41.06 ± 11.70 mg/mL), which was 32.93% higher compared to the group of animals receiving MWCNT at the concentration of 0.25 mg/mL (27.54 ± 5.83 mg/mL) (Figure 4).

The study showed statistically significant differences in TP in the liver of rats receiving MWCNTs at a concentration of 0.5 mg/mL and rats in the control group and exposed to 18 months of exposure (41.06 ± 11.70 and 31.54 ± 4.34 mg/mL, respectively) compared to the groups exposed to the same concentrations and euthanized after 12 months (27.20 ± 3.10 and 24.72 ± 5.01 mg/mL, respectively). TP levels in the euthanized groups after 18 months were 33.76% and 21.62% higher, respectively, compared to the 12-month exposure (Figure 4). 

##### Kidneys

The analysis showed statistically significant differences in renal TP levels in rats given MWCNTs at 0.5 mg/mL after 12-month exposure, compared to animals exposed to 0.25 mg/mL (19.72 ± 0.56 and 24.12 ± 2.71 mg/mL, respectively); there was an 18.42% reduction in TP levels (Figure 4). In contrast, there were no significant differences in TP levels in the kidneys of rats given MWCNTs at concentrations of 0.03 mg/mL (21.52 ± 2.98 mg/mL), 0.25 mg/mL (24.12 ± 2.71 mg/mL) and 0.5 mg/mL (19.72 ± 0.56 mg/mL) compared to the control group (24.87 ± 6.69 mg/mL) (Figure 4). 

The mean TP levels in the kidneys of rats were significantly higher (by 15.33%) in the group of animals given MWCNTs at a concentration of 0.5 mg/mL and euthanized after 18 months (27.33 ± 5.70 mg/mL) compared to 12 months (19.72 ± 0.56 mg/mL) (Figure 4). 

#### 3.3.2. Reduced Glutathione (GSH)

##### Serum

The research showed statistically significant differences in serum reduced glutathione (GSH) levels in rats given MWCNTs at a concentration of 0.25 mg/mL and euthanized after 12 months (11.61 ± 5.47 nmol/mg protein) compared to a group of animals given MWCNT at a concentration of 0.03 mg/mL (4.58 ± 1.52 mg/mL). GSH levels in this group were 60.55% higher (Table 1).

There was a statistically significant reduction of 62.19% in serum GSH levels in animals exposed to MWCNT at a concentration of 0.5 mg/mL (4.39 ± 2.74 nmol/mg protein) compared to animals given nanotubes at a concentration of 0.25 mg/mL (11.61 ± 5.47 nmol/mg protein) and exposed for 12 months (Table 1).

The research showed statistically significant differences in serum GSH levels in rats receiving MWCNTs at a concentration of 0.5 mg/mL and euthanized after 18 months (13.49 ± 8.59 nmol/mg protein) compared to a group of animals receiving nanotubes at the same concentration and euthanized after 12 months (4.39 ± 2.74 nmol/mg protein). GSH levels in this group were 67.46% higher compared to the group of animals euthanized after 12 months (Table 1). 

The research also showed a statistically significant difference in serum GSH levels in the control group euthanized after 18 months (4.39 ± 2.13 nmol/mg protein) compared to the control group euthanized after 12 months (9.54 ± 3.91 nmol/mg protein). GSH levels in this group were 53.98% lower compared to the group of animals euthanized after 12 months (Table 1). 

##### Kidneys

The research showed a significant difference in GSH levels in the kidneys of the control animals. Significantly higher GSH levels, by 32.86%, in the kidney were recorded for the 18-month exposure group compared to the 12-month exposure (11.90 ± 1.89 and 7.99 ± 3.46 nM/mg protein, respectively) (Figure 5).

#### 3.3.3. Thiobarbituric Acid Reactive Substances (TBARS) 

##### Kidneys

The mean values of TBARS levels in the kidneys of the rats in different study groups exposed for 18 months compared to the 12-month exposure varied but were not statistically significant (Figure 6). In contrast, TBARS levels were found to be significantly lower in the control group after 18 months (0.87 ± 0.22 nM MDA/mg protein) compared to the 12-month exposure (2.70 ± 1.70 nM MDA/mg protein). TBARS levels were 67.78% lower (Figure 6).

#### 3.3.4. TEAC (Trolox Equivalent Antioxidant Capacity)

##### Serum

The research showed there was a statistically significant difference in serum TEAC capacity in rats given the nanotube solution at a concentration of 0.25 mg/mL and euthanized after 18 months (71.36 ± 25.14 nmol/mg protein) compared to the group of animals receiving nanotubes at the same concentration and euthanized after 12 months (40.83 ± 13.66 nmol/mg protein) (Table 1). The TEAC capacity value in this group was 42.78% higher compared to the group of animals euthanized after 18 months (Table 1).

##### Liver

Studies showed statistically significant differences in TEAC in the liver of rats receiving MWCNTs at a concentration of 0.25 mg/mL compared to the animals receiving nanotubes at a concentration of 0.03 mg/mL (20.23 ± 4.02 and 15.04 ± 1.75 nmol/mg protein, respectively) after the 18-month exposure; TEAC levels in the group receiving nanotubes at a concentration of 0.25 mg/mL were 25.65% higher (Figure 7).

Additionally, in the group of rats given nanotubes at a concentration of 0.5 mg/mL, statistically significant differences were shown when compared to the rats exposed for 18 months to the concentration of 0.25 mg/mL (12.99 ± 3.11 and 20.13 ± 4.02 nmol/mg protein, respectively), TEAC levels in rats receiving nanotubes at a concentration of 0.5 mg/mL were 35.79% lower (Figure 7).

TEAC was significantly lower in the liver of the rats that received MWCNTs at a concentration of 0.5 mg/mL and in the control group when euthanized after 18 months (12.99 ± 3.11 and 16.99 ± 1.95 nmol/mg protein, respectively) compared to the groups with the same concentrations but euthanized after 12 months (21.55 ± 3.04 and 20.83 ± 3.25 nmol/mg protein, respectively). TEAC levels after the 18-month exposure were 39.72% and 18.43% lower, respectively, compared to the groups euthanized after 12 months (Figure 7).

#### 3.3.5. Nitric Oxide (*NO*)

##### Liver

There were statistically significant differences in NO in the livers of rats receiving MWCNTs at a concentration of 0.03 mg/mL and euthanized after 18 months (0.23 ± 0.05 nmol/mg protein) compared to a group of rats with the same concentration but euthanized after 12 months (0.31 ± 0.07 nmol/mg protein). NO concentration after 18-month exposure was 25.81% lower compared to the euthanized group after 12 months (Figure 8). 

##### Kidneys

The research showed that there was a statistically significant difference in NO levels in the kidneys of rats treated with MWCNTs at 0.5 mg/mL compared to the control animals (0.12 ± 0.02 and 0.26 ± 0.04 nM/mg protein, respectively); there was a 53.85% increase in NO levels during the 18-month exposure (Figure 8). 

When comparing exposure times, the research showed significantly higher NO levels in the kidneys of rats given MWCNTs at a concentration of 0.25 mg/mL and euthanized after 18 months (0.18 ± 0.05 nM/mg protein) when compared to 12-month exposure (0.11 ± 0.02 nM/mg protein). NO levels were 38.89% higher (Figure 8). A similar comparison also showed that the mean NO levels in the kidneys of rats in the control group euthanized after the 18 months were significantly higher compared to the 12-month exposure (0.26 ± 0.04 and 0.12 ± 0.05 nM/ mg protein, respectively). NO levels were 53.85% higher (Figure 8).

#### 3.3.6. Glutathione S-Transferase (GST)

##### Serum

The research showed statistically significant differences in serum GST levels in rats given MWCNTs at a concentration of 0.25 mg/mL and euthanized after 18 months (7.51 ± 3.35 nmol/min/mg protein) compared to a group of animals receiving nanotubes at a concentration of 0.03 mg/mL (3.46 ± 1.39 nmol/min/mg protein) and the control group (3.08 ± 2.03 nmol/min/mg protein) euthanized after 18 months. The value of GST activity in this group was 53.93% higher compared to the group of animals receiving nanotubes at a concentration of 0.03 mg/mL and 58.99% higher compared to the control group (Table 1).

Additionally, the research showed a statistically significant difference in serum glutathione s-transferase activity in rats given MWCNTs at a concentration of 0.25 mg/mL and euthanized after 18 months (7.51 ± 3.35 nmol/min/mg protein) compared to the group of animals given nanotubes at a concentration of 0.25 mg/mL and euthanized after 12 months (3.27 ± 2.26 nmol/min/mg protein). The GST activity value in this group was 56.46% higher compared to the group of animals that were euthanized after 12 months (Table 1). 

##### Kidneys

The analysis showed statistically significant differences in GST activity in the kidneys of rats given a 0.5 mg/mL nanotube solution after 18-month exposure compared to animals exposed to a concentration of 0.25 mg/mL (16.95 ± 6.16 and 41.89 ± 28.32 nM/min/mg protein, respectively); there was a 59.54% reduction in glutathione S-transferase activity (Figure 9). GST activity was significantly higher in the kidneys of rats given MWCNTs at a concentration of 0.25 mg/mL and euthanized after 18 months compared to the 12-month exposure (23.44 ± 4.81 and 41.89 ± 28.32 nM/min/mg protein, respectively). GST activity increased by 44.04% (Figure 9).

#### 3.3.7. Catalase (CAT)

Catalase levels between all groups given MWCNTs at three doses: 0.03 mg/mL, 0.25 mg/mL, and 0.5 mg/mL and at two times of exposure showed no statistically significant differences (Figure 10).

## 4. Discussion

With the increasing use of nanoparticles in diagnostics and treatment, there are concerns about their toxic effects on the human body [63]. Due to their very small size, NPs are able to enter the human body through a variety of routes by which they can interact with intracellular structures and macromolecules for long periods of time [63]. The toxicity of nanomaterials is mainly due to the production of excess reactive oxygen species (ROS) [64]. Oxidative stress biomarkers are, therefore, important tools for assessing nanoparticle toxicity. Our study objectives were to evaluate selected biochemical parameters reflecting the intensity of oxidative stress in serum, liver, and kidney tissue in animals after 12 and 18 months of administration with a single injection to the knee joint with a solution of multi-walled carbon nanotubes.

The search for new methods of treating articular cartilage damage is a challenge for scientists. Degenerative changes resulting from damage to the articular cartilage are the most common disease of the musculoskeletal system. Osteoarthritis is currently the third most common cause of disability in the world population. The latest research focuses on obtaining and detailed characterization of a new biomaterial with the potential to repair damage to joint cartilage. The creation of an appropriate biomaterial used for cell transplantation may contribute to the development of nanomedicine and regenerative medicine [3]. However, before this happens, it is necessary to characterize the full toxicological profile of the new materials, including their impact on the prooxidant–antioxidant balance Therefore, for the purposes of our research, previously developed, novel carbon nanotubes were administered to one of the knee joint of the right hind leg. The second “clean” joint served as a control to observe possible changes in the joint to which the materials were administered.

Nanomaterials introduced into biological systems are immediately coated by proteins in vivo. They induce oxidative stress on adsorbed proteins, and hence they alter the protein structures, which determines the fate pathways and biological impacts of nanomaterials. Carbon nanotubes (CNTs) have been suggested to cause protein oxidation [65]. Clichici et al., investigating the effect of MWCNTs in Wistar rats, observed a slight increase in plasma protein levels 3 h after a single dose (1.5 mL) of a 270 mg/L solution of ss-DNA-MWCNTs; however, this difference was not statistically significant when compared to the control groups [66]. In our study, after 12 months, an increase in rat serum total protein levels dependent on the concentration of nanotubes was observed; however, only for the highest concentration of 0.5 mg/mL, a statistically significant difference (33.84% increase) was shown compared to the control group. For the 18-month exposure period, there was also a statistically significant increase of 45.57% in serum TP levels in animals given nanotubes at a concentration of 0.5 mg/mL compared to the animals given nanotubes at a concentration of 0.25 mg/mL. In studies of the cellular influence of carbon nanotubes cellular on blood composition, oxidized carbon nanotubes (CNTox) and doxorubicin-functionalized carbon nanotubes (CNT-Dox) and doxorubicin slightly increased the level of total protein in the blood of experimental animals [67]. Our study showed a statistically significant difference in TP levels in the liver of animals after 18 months as compared to the 12-month exposure. TP levels were 32.93% higher in rats given MWCNTs at a concentration of 0.5 mg/mL compared to a concentration of 0.25 mg/mL. For the other groups, TP levels were not significantly different. In our study, TP levels in the kidneys of rats were comparable between groups; however, TP increased proportionally to MWCNTs concentration within groups of animals euthanized after 18 months. 

Glutathione (GSH) is a thiol that plays a major role in maintaining the balance between free radicals and antioxidants in the body. The sulfhydryl (-SH) groups of cysteine, which are involved in reduction and conjugation reactions, are responsible for this effect. These transformations enable the removal of peroxides and many xenobiotic compounds [68]. According to a study by Adedara et al., the administration of a solution of carboxylated MWCNTs to Wistar rats resulted in a dose-dependent decrease in serum GSH levels. Glutathione levels decreased by 29% at a nanotube dose of 0.25 mg/kg, 41% at a dose of 0.5 mg/kg, 74% at a dose of 0.75 mg/kg, and 84% at a dose of 1 mg/kg, respectively, compared to the control group [35]. Clichici et al., investigating the effect of MWCNTs in adult male Wistar rats, observed a 2-fold decrease in plasma GSH levels at 6 h and 2.25-fold at 48 h after administration of a single amount—1.5 mL of ss-DNA-MWCNTs at a concentration of 270 mg/L [66]. Reddy et al., in their study, demonstrated a dose-dependent decrease in plasma GSH levels in Wistar strain rats compared to the control group [69]. The animals received the MWCNTs solution intratracheally at a dose of 0.2 mg/kg bw, 1 mg/kg bw, and 5 mg/kg bw, sequentially. At the same time, glutathione concentrations decreased by about 70% in rats 24 h after administration of the MWCNTs solution and gradually increased after 1 week, 1 month, and 3 months after exposure [69]. Our study after 12 months showed a statistically significant reduction by 62.19% in serum GSH levels in rats exposed to nanotubes at a concentration of 0.5 mg/mL compared to a concentration of 0.25 mg/mL. 

Awogbindin et al. observed a statistically significant about 60% lower GSH levels in the livers of animals receiving MWCNTs at a dose of 1 mg/kg body weight than in control animals [70]. In the study by Fang et al., mice were administered MWCNTs at a dose of 100 mg/kg bw and they were euthanized after 5 days. The experiment showed a 5% reduction in GSH levels in the liver of animals given carbon nanotubes compared to the control group [71]. Yang et al., in their study, investigated the accumulation and toxicity of SWCNTs administered intravenously to mice. The animals were administered SWCNTs at doses of 40 µg/kg bw, 200 µg/kg bw, and 1 mg/kg bw and were euthanized after 90 days. The study showed reduced glutathione levels in the livers of animals in all groups that were given carbon nanotubes: 90, 89, and 70 mg/g protein, respectively, compared to the GSH levels in the control group of 95 mg/g protein [72]. Glutathione acts as a reducing equivalent, helping to neutralize the damaging effects of electrolytes, free radicals, and oxidants, as well as enabling the detoxification of xenobiotics. Low GSH levels after nanotube administration may suggest that significant amounts of this molecule have been utilized, probably in processes involving scavenging of reactive oxygen species and/or coupling of CNT. In our study, we did not observe statistically significant differences in GSH in rat liver and kidney.

The first line of defense against reactive oxygen species is antioxidant enzymes, which include glutathione s-transferase (GST) [73]. The effect of GST consists of being a catalyst in the reaction of coupling glutathione with various types of xenobiotics. As a result, their reactivity decreases and their solubility in water increases, which promotes their elimination from the body. Glutathione transferases are primarily responsible for the metabolism of exogenous compounds, especially carcinogens, and the detoxification of potentially harmful endogenous substances [74,75,76]. S-transferases protect DNA molecules and cell membranes from free radicals [77]. In our study, the 18-month exposure resulted in a statistically significant increase in GST activity—by 53.93% in the group of rats exposed to nanotubes at a concentration of 0.25 mg/mL, compared to animals receiving MWCNTs at a concentration of 0.03 mg/mL and by 58.99% compared to the control group. There was a 56.46% increase in serum GST activity in a group of rats receiving MWCNTs at a concentration of 0.25 mg/mL after an 18-month exposure compared to a 12-month exposure period. GST for the other concentrations was similar for both exposure periods. 

In their study, HelmyAbdou et al. wanted to gain insight into new treatment options for the hepatotoxic effects of MWCNTs exposure [78]. Carbon nanotubes were administered to rats at a dose of 1 g/kg body weight, and it was observed that glutathione s-transferase activity levels decreased by 47% [78]. In a study by Awogbindin et al., GST levels in a group of animals given MWCNTs at a dose of 1 mg/kg bw decreased by approximately 50% compared to the control group [70]. Adedara et al., in a study of 8-week-old Wistar rats exposed to MWCNTs, observed a dose-dependent decrease in liver GST activity of 25%, 43%, 81%, and 96% at nanotube doses of 0.25 mg/kg bw, 0.5 mg/kg bw, 0.75 mg/kg bw, and 1 mg/kg bw, sequentially, compared to the control group [35]. In our study, there was no correlation in liver GST levels in the groups given different concentrations of nanotubes. 

A study by Lim et al. focused on the potential adverse effects of multi-walled carbon nanotubes (MWCNTs) on pregnant female rats and embryonic development following maternal exposure [79]. Pregnant female rats were orally administered nanotubes at doses of 0, 8, 40, 200, and 1000 mg/kg bw/day. GST levels in the kidneys of the nanotube-exposed animals were lower than in the control group, and the lowest concentrations were recorded after administration of 1000 mg/kg bw/day. This value was 16.18% lower [79]. Adedara et al. demonstrated a dose-dependent decrease in GST activity in the kidneys of MWCNT-exposed animals by an average of 57% [35]. In our study, the mean glutathione S-transferase activity in rat kidneys was similar between the groups. 

The total Trolox equivalent antioxidant capacity test was designed to determine the overall antioxidant power of samples together with the antioxidant and their interaction. TEAC assessment in body fluid has been used as one of the biological markers to monitor oxidative stress [80]. Reddy et al., in their study, showed, following intratracheal administration, an MWCNT dose-dependent (0.2, 1.0, and 5.0 mg/kg bw), slight decrease in the total antioxidant capacity in the plasma of rats compared to the control group [69]. In our study, the group receiving MWCNT at a concentration of 0.5 mg/mL and the control group showed a statistically significant difference in TEAC levels after 18 months of exposure. Moreover, our study shows a 35.79% decrease observed in rats receiving nanotubes at a concentration of 0.5 mg/mL compared to the 0.25 mg/mL group. The reduced Trolox equivalent antioxidant capacity may indicate reduced antioxidant protection in the livers of rats. The Trolox equivalent antioxidant capacity in the kidneys of rats varied, but no statistically significant differences were found. 

Numerous reports indicate that increased concentration of MDA and other products of peroxidation of thiobarbituric acid reactive substances (TBARS) is an important marker of oxidative stress in biological material [81]. In a study by Adedara et al. conducted on Wistar rats exposed to MWCNTs, a dose-dependent significant increase in hepatic levels of MDA was observed. The percentage increase in hydrogen peroxide (H_2_O_2_) levels was: 31%, 36%, 39%, and 44%, sequentially, with increased levels of lipid peroxidation by 32%, 51%, 64%, and 66%, sequentially, at doses of 0.25 mg/kg bw, 0.5 mg/kg bw, 0.75 mg/kg bw, 1.0 mg/kg bw, respectively, compared to the control group [35]. Clichici et al., investigating the effect of MWCNTs in rats, observed a significant increase in MDA levels in serum 1 h after administration of the ss-DNA-MWCNTs complex at a concentration of 270 mg/L, compared to the control group. These differences persisted for 3 and 6 h after administration of the ss-DNA-MWCNTs complex, while no statistically significant difference was observed after 144 h compared to the control group [66]. Our study showed no statistically significant differences in serum lipid peroxidation product levels in rats given MWCNTs at different concentrations compared to the control group. The concentration of TBARS is an important marker of the intensity of the lipid peroxidation process. When the ROS react with cellular macromolecules, they enhance the process of lipid peroxidation, cause cell damage, induce protein and nucleic acid modifications, and can lead to, ultimately, organ dysfunction [81]. Awogbindin et al. observed a significant 50% increase in malondialdehyde and reactive oxygen and nitrogen species (RONS) levels in the liver of rats that were administered intraperitoneally MWCNTs at a dose of 1 mg/kg bw compared to the control group [70]. Reddy et al. assessed oxidative stress and antioxidant status in rat serum following the administration of MWCNTs produced using an electric arc [69]. The rats were administered MWCNTs at doses of 0.2, 1, and 5 mg/kg bw, and blood samples were collected on days 1, 7, 30, and 90 after exposure. According to the study, lipid peroxidation increased with the increase in concentrations of carbon nanotubes and also increased during the first week of exposure, compared to day 1 of exposure, and TBARS levels decreased at 30 and 90 days after application [69]. TBARS represents the manifestation of ROS that have managed to modify plasma or cell membrane lipids, yielding intermediates such as MDA. In 12-month-old animals, results higher than in the control group may indicate that exposure to MWCNTs induces oxidative stress by reducing the total antioxidant capacity in rats. Awogbindin et al. observed a significant increase in malondialdehyde and reactive oxygen and nitrogen species (RONS) levels in the kidneys of rats that were administered MWCNTs intraperitoneally at a dose of 1 mg/kg bw; lipid peroxidation levels in both the liver and kidney of animals in this group were increased by 50% compared to the control group. Our research has shown that TBARS levels in kidneys varied after 12 and 18 months from exposure, but no statistically significant differences were found [70]. 

Catalase is one of the better-studied enzymes that plays a key role in protecting cells from the toxic effects of hydrogen peroxide [82]. CAT protects hemoglobin by removing more than half of the hydrogen peroxide produced in erythrocytes exposed to significant oxygen concentrations [83]. The role of catalase in antioxidant defense depends on the tissue type and the pattern of oxidant-mediated tissue damage [84]. Wang et al. investigated oxidative stress in the liver of rats after administration of MWCNTs and SWCNT, among others, and observed a 68.06 ± 1.23% reduction in CAT levels compared to the control group [85]. In a study by Awogbindin et al., CAT levels in the liver of rats given MWCNTs at a concentration of 1 mg/kg bw decreased by 75% compared to the control group [70]. The study by Adedara et al. focused on MWCNT-induced hepatotoxicity. Rats were given MWCNTs at doses of 025, 0.5, 0.75, and 1 mg/kg bw for 5 consecutive days [35]. Liver catalase levels in the MWCNT-exposed groups increased by 31%, 37%, 46%, and 49%, respectively, compared to the control group [35]. Catalase also is crucial against oxidative damage caused by ROS by degrading surplus H202 [86]. The decomposition of hydrogen peroxide takes place mainly with the participation of GPx, so catalase has only auxiliary functions and shows activity in the event of extremely high H_2_O_2_ concentrations [86]. Our study has shown that catalase activity in the liver and kidney was comparable within the test and control groups of animals euthanized after 12 and 18 months, respectively [62]. 

The production of reactive oxygen species (ROS) is important in both normal physiology and the pathogenesis of many diseases. ROS include partially reduced forms of molecular oxygen, such as hydroxyl radical (^•^OH), superoxide anion (O_2_^•−^), hydrogen peroxide (H_2_O_2_), lipid peroxides, and hypochlorous acid (HClO). The accumulation of ROS can be accompanied by the production of reactive nitrogen species (RNS), such as the highly reactive peroxynitrite anion, a strong oxidant formed in the O_2_^•−^ reaction, and nitric oxide (NO) [87]. Substantial evidence suggests that nitric oxide (NO) induces nitrosative stress, thereby leading to cell damage through direct or indirect interaction with biomolecules, especially when the protective antioxidant system is depleted [88]. Nitric oxide is a reactive form of nitrogen and is important in the redox biology of hepatocytes [89]. NO plays an important and diverse, both cytoprotective and cytotoxic, role in the liver. Factors determining whether nitric oxide will have a protective or detrimental effect are the location of NO production, the amount of NO produced, and the relative amounts of superoxide anion produced at the same site as nitric oxide [90]. In a study by Adedara et al. in animals treated with MWCNTs at a concentration of 0.25, 0.5, 0.75, and 1 mg/kg for 5 consecutive days, liver NO levels increased by 19%, 63%, 75%, and 80%, respectively, compared to the control group [35]. Awogbindin et al. investigated the mitigating effect of kolaviron on hepatorenal damage in rats exposed to the group receiving MWCNTs at a dose of 1 mg/kg body weight; a significant difference was noted, the NO levels in the liver of the nanotube-exposed animals were approximately 60% higher than that of the control group [70]. Adedara et al. have shown that in rats given MWCNT at a dose of 1 mg/kg bw, an increase in NO levels in the kidneys of approximately 25% was observed compared to the control group [35]. Our study showed significant differences in the NO levels in the kidneys of rats (38.89% increase) that received MWCNTs at a concentration of 0.25 mg/mL after 18 months, compared to the 12-month exposure. NO levels were 38.89% higher. Longer exposure time to the carbon nanotube solution results in higher NO levels in the rats’ kidneys.

Studies have shown that MWCNTs may increase ROS production in various cell types, such as macrophages, fibroblasts, and epithelial cells [55]. The results of our research also confirm this. Although the introduction of nanotubes into a living organism causes an increase in oxidative stress, for some parameters, it is a tendency that reverses after prolonged exposure as the body begins to adapt to the changes taking place in it. Therefore, it is very important to modify nanotubes properly when thinking about future applications in tissue engineering, such as combining them with antioxidants, which will help to reduce their toxicity.

## 5. Conclusions

Increased liver TP levels in rats given nanotubes suggest a proportional relationship between MWCNTs concentration and toxicity.Our study, after 12 months, showed a statistically significant reduction of 62.19% in serum GSH levels in rats exposed to nanotubes at higher concentrations compared to lower concentrations.The 18-month exposure resulted in a statistically significant increase in GST activity—by 53.93% in the group of rats exposed to nanotubes at higher concentrations compared to animals receiving MWCNTs at lower concentrations and by 58.99% compared to the control group. However, after the next 6 months of the observation, GST levels were lower. This may suggest the reduced efficiency of repair processes in the process of aging of the animals exposed to higher concentrations of carbon nanotubes, leading to GST depletion.Moreover, a 35.79% decrease was observed in rats receiving nanotubes at higher concentrations compared to the group receiving lower concentrations of MWCNTs. Lower TEAC may indicate reduced antioxidant protection in the liver of rats exposed to higher concentrations of MWCNTs.Longer exposure time to the carbon nanotube solution results in higher NO levels in the rats’ kidneys.

## 6. Summary

There is a link between nanotoxicity and excessive oxidative stress, especially in relation to the liver and kidneys, which are susceptible to oxidative damage. An analysis of oxidative stress parameters can be a key indicator of the toxic potential of multi-walled carbon nanotubes (MWCNTs), among others. There are a large number of factors influencing the toxic potential of MWCNTs, including the generation of oxidative stress, which often complicates the interpretation of results and the comparison of different studies. It is certainly important to standardize the characterization of multi-walled carbon nanotubes.

## Figures and Tables

**Figure 1 antioxidants-12-00464-f001:**
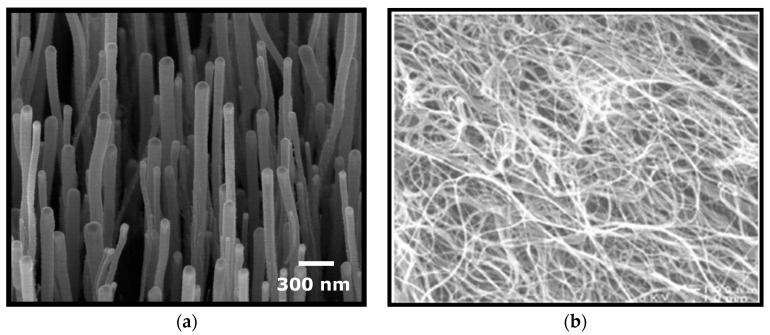
Examples of TEM images on multi-walled carbon nanotubes (MWCNT) used in our experiments. (**a**) MWCNT growth on a Si plate using Fe as a catalyst; (**b**) MWCNT in (spaghetti-like) growth process using Fe as a catalyst.

**Figure 2 antioxidants-12-00464-f002:**
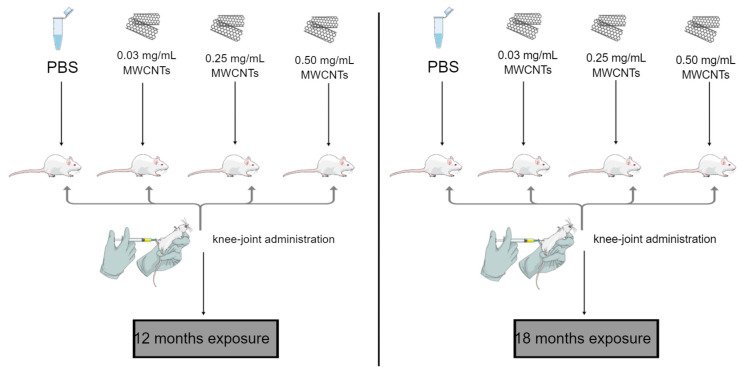
Schematic representation of the designed experiment.

**Figure 3 antioxidants-12-00464-f003:**
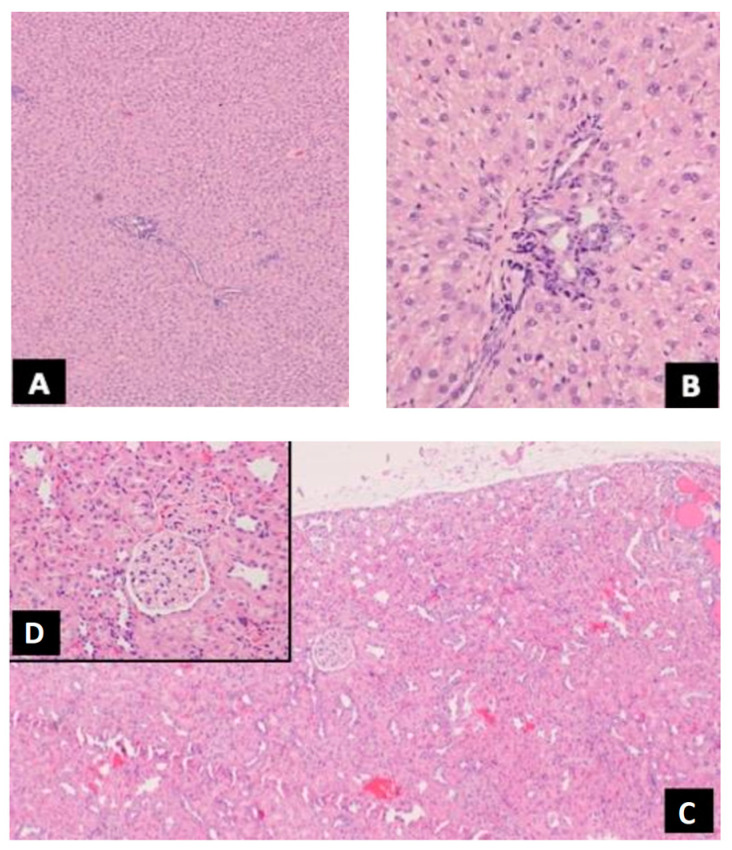
(**A**) Fragment of liver with very well-preserved structure; (**B**) inset: closer view of the peripheral part of liver lobule with delicate (slight) features of small droplets steatosis. Primary objective magnification: A, 4×, B, 10×.; (**C**) Fragment of kidney with very well-preserved structure. There are some tubules with widened lumen; (**D**) inset: closer view of glomerulus without any significant changes. Primary objective magnification: A, 4×, B, 10×.

**Figure 4 antioxidants-12-00464-f004:**
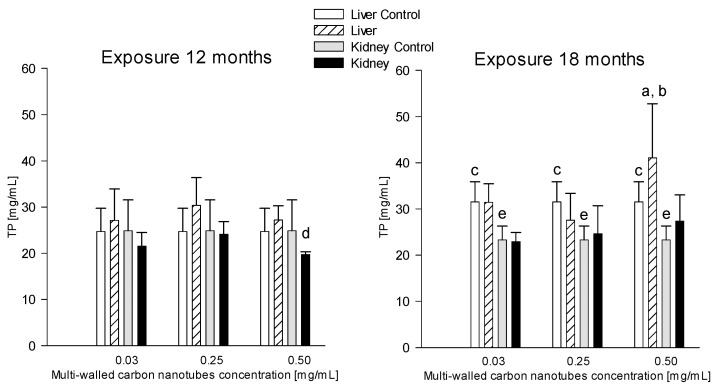
Comparison of total protein (TP) concentrations in the liver and kidneys of animals exposed to multi-walled carbon nanotubes for 12 and 18 months. (a)—statistically significant difference for the group receiving MWCNTs in concentration of 0.5 mg/mL versus 0.25 mg/mL in the livers (*p* = 0.022493); (b)—statistically significant difference for the group receiving MWCNTs in concentration of 0.5 mg/mL at 18 months versus 12 months in the livers (*p* = 0.032701); (c)—statistically significant difference for control group at 18 months versus control at 12 months in the livers (*p* = 0.030844); (d)—statistically significant difference for group receiving MWCNTs in concentration of 0.5 mg/mL versus 0.25 mg/mL in the kidneys (*p* = 0.0197); (e)—statistically significant difference for control group at 18 months versus control at 12 months in the kidneys (*p* = 0.008621).

**Figure 5 antioxidants-12-00464-f005:**
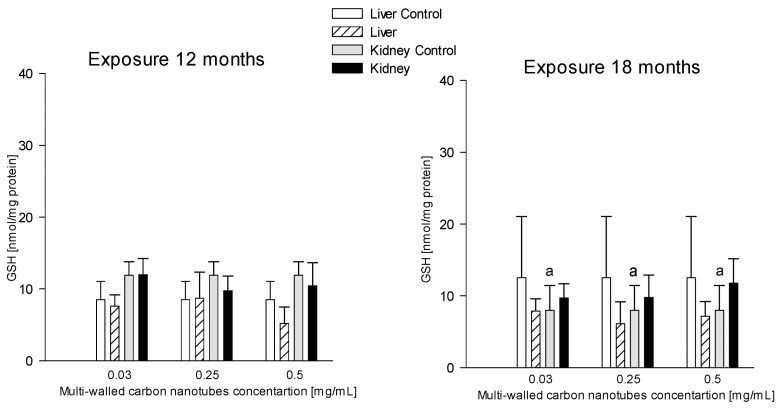
Comparison of reduced glutathione (GSH) concentrations in the liver and kidneys of animals exposed to multi-walled carbon nanotubes for 12 and 18 months. (a)—statistically significant difference for control group at 18 months versus control at 12 months in the kidneys (*p* = 0.035571).

**Figure 6 antioxidants-12-00464-f006:**
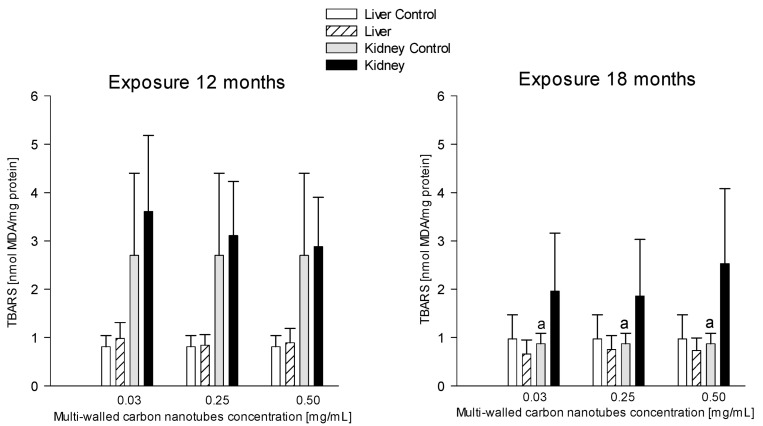
Comparison of thiobarbituric acid reactive substances (TBARS) concentrations in the liver and kidneys of animals exposed to multi-walled carbon nanotubes for 12 and 18 months. (a)—statistically significant difference for control group at 18 months versus control at 12 months (*p* = 0.025661).

**Figure 7 antioxidants-12-00464-f007:**
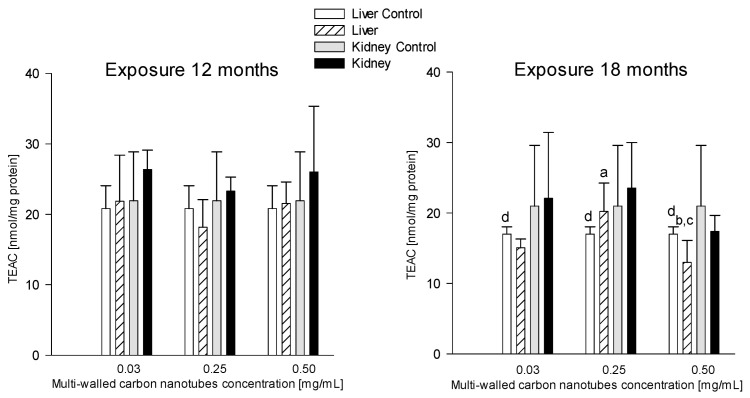
Comparison of Trolox equivalent antioxidant capacity (TEAC) concentrations in the liver and kidneys of animals exposed to multi-walled carbon nanotubes for 12 and 18 months. (a)—statistically significant difference for the group receiving MWCNTs in a concentration of 0.25 mg/mL versus 0.03 mg/mL in the livers (*p* = 0.024330); (b)—statistically significant difference for the group receiving MWCNTs in a concentration of 0.5 mg/mL versus 0.25 mg/mL in the livers (*p* = 0.001632); (c)—statistically significant difference for the group receiving MWCNTs in nanotube concentration of 0.5 mg/mL at 18 months versus 12 months in the livers (*p* = 0.000702); (d)—statistically significant difference for control group at 18 months versus control at 12 months in the livers (*p* = 0.037120).

**Figure 8 antioxidants-12-00464-f008:**
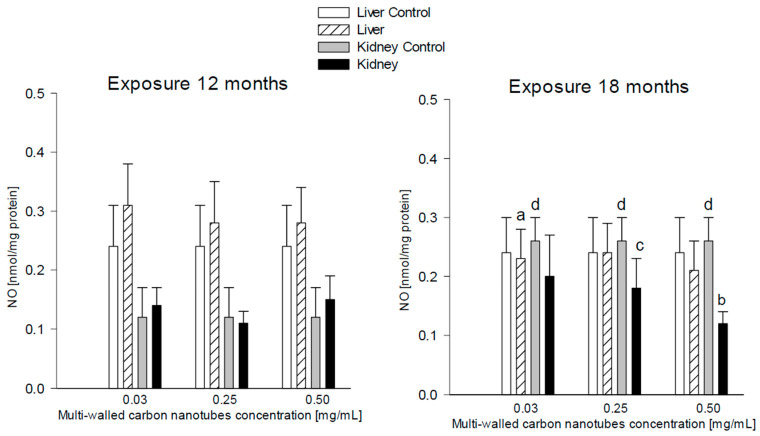
Comparison of nitric oxide (NO) concentrations in the liver and kidneys of animals exposed to multi-walled carbon nanotubes for 12 and 18 months. (a)—statistically significant difference for the group receiving MWCNTs in a concentration of 0.03 mg/mL at 18 months versus 12 months in the livers (*p* = 0.051588); (b)—statistically significant difference for MWCNTs in a concentration of 0.5 mg/mL versus control group in the kidneys (*p* = 0.0049); (c)—statistically significant difference for the group receiving MWCNTs in a concentration of 0.25 mg/mL at 18 months versus 12 months in the kidneys (*p* = 0.008052); (d)—statistically significant difference for the control group at 18 months versus control at 12 months in the kidneys (*p* = 0.000269).

**Figure 9 antioxidants-12-00464-f009:**
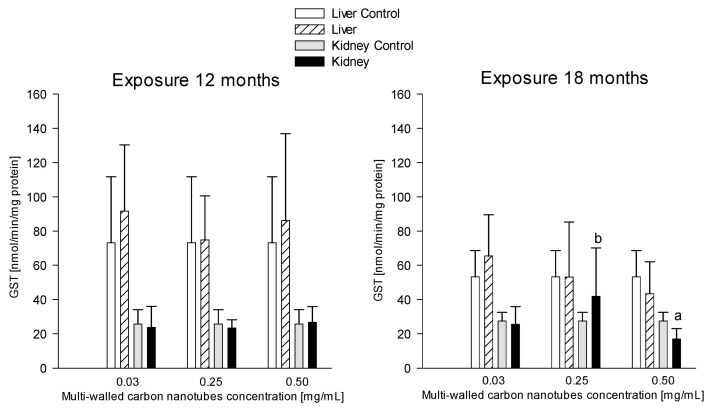
Comparison of glutathione S-transeferase (GST) concentrations in the liver and kidneys of animals exposed to multi-walled carbon nanotubes for 12 and 18 months. (a)—statistically significant difference for MWCNTs in concentration of 0.5 mg/mL versus 0.25 mg/mL in the kidneys (*p* = 0.0178), (b)—statistically significant difference for MWCNTs in concentration of 0.25 mg/mL at 18 months versus 12 months in the kidneys (*p* = 0.041126).

**Figure 10 antioxidants-12-00464-f010:**
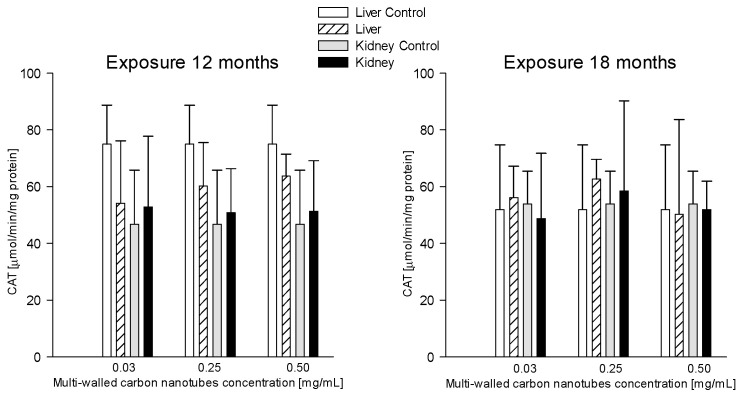
Comparison of catalase (CAT) concentrations in the liver and kidneys of animals exposed to multi-walled carbon nanotubes for 12 and 18 months.

**Table 1 antioxidants-12-00464-t001:** Concentration of selected parameters of oxidative stress in serum in animals exposed to multi-walled carbon nanotubes after 12 and 18 months. (a)—statistically significant difference for control group at 18 months versus control at 12 months (*p* = 0.040649); (b)—statistically significant difference in group receiving MWCNTs in concentration of 0.25 mg/mL versus 0.03 mg/mL (*p* = 0.09144); (c)—statistically significant difference in group receiving MWCNTs in concentration of 0.5 mg/mL versus 0.25 mg/mL (*p* = 0.01629); (d)—statistically significant difference in group receiving MWCNTs in concentration of 0.25 mg/mL versus control group (*p* = 0.02529); (e)—statistically significant difference in group receiving MWCNTs in concentration of 0.25 mg/mL versus 0.03 mg/mL (*p* = 0.017977); (f)—statistically significant difference in group receiving MWCNTs in concentration of 0.5 mg/mL versus 0.25 mg/mL (*p* = 0.014816); (g)— statistically significant difference in group receiving MWCNTs in concentration of 0.25 mg/mL versus 0.03 mg/mL (*p* = 0.031157); (h)—statistically significant difference in group receiving MWCNTs in concentration of 0.25 mg/mL versus control group (*p* = 0.016943); (i)—statistically significant difference in group receiving MWCNTs in concentration of 0.25 mg/mL at 18 months versus 12 months (*p* = 0.027652); (j)—statistically significant difference in group receiving MWCNTs in concentration of 0.25 mg/mL at 18 months versus 12 months (*p* = 0.025886); (k)—statistically significant difference in group receiving MWCNTs in concentration of 0.5 mg/mL versus control group (*p* = 0.029093); (l)—statistically significant difference in group receiving MWCNTs in concentration of 0.5 mg/mL versus 0.25 mg/mL (*p* = 0.01629); (m)—statistically significant difference in group receiving MWCNTs in concentration of 0.5 mg/mL versus 0.25 mg/mL (*p* = 0.014816); (n)—statistically significant difference for control group at 18 months versus control at 12 months (*p* = 0.040649); (o)—statistically significant difference in group receiving MWCNTs in concentration of 0.5 mg/mL at 18 months versus 12 months (*p* = 0.01782). Mean (n = 6) ±SD.

Concentration of Multi-Walled Carbon Nanotubes [mg/mL]	Parameter	Control after 12 Months	After 12 Months Exposure	Control after 18 Months	After 18 Months Exposure
0.03	TP [mg/mL]	70.50 ± 17.38	86.59 ± 12.51	88.32 ± 28.33	94.16 ± 19.92
	GSH [nmol/mg protein]	9.54 ± 3.91	4.58 ± 1.52	4.39 ± 2.13 ^a^	6.31 ± 6.50
	GST [nmol/min/mg protein]	3.70 ± 1.12	4.63 ± 2.26	3.08 ± 2.03	3.46 ± 1.39
	TEAC [nmol/mg protein]	49.25 ± 23.16	39.82 ± 7.41	42.25 ± 16.73	50.74 ± 23.93
	TBARS [nmol MDA/mg protein]	0.66 ± 0.14	0.54 ± 0.06	0.45 ± 0.21	0.53 ± 0.23
	NO [nmol/mg protein]	0.49 ± 0.26	0.38 ± 0.18	0.47 ± 0.17	0.52 ± 0.30
0.25	TP [mg/mL]	70.50 ± 17.38	86.68 ± 26.45	88.32 ± 28.33	49,32 ± 18.57 ^b,c,d^
	GSH [nmol/mg protein]	9.54 ± 3.91	11.61 ± 5.47 ^e^	4.39 ± 2.13 ^f^	10.60 ± 3.90
	GST [nmol/min/mg protein]	3.70 ± 1.12	3.27 ± 2.26	3.08 ± 2.03	7.51 ± 3.35 ^g,h,i^
	TEAC [nmol/mg protein]	49.25 ± 23.16	40.83 ± 13.66	42.25 ± 16.73	71.36 ± 25.14 ^j^
	TBARS [nmol MDA/mg protein]	0.66 ± 0.14	0.69 ± 0.35	0.45 ± 0.21	0.71 ± 0.36
	NO [nmol/mg protein]	0.49 ± 0.26	0.45 ± 0.17	0.47 ± 0.17	0.65 ± 0.40
0.5	TP [mg/mL]	70.50 ± 17.38	94.36 ± 9.55 ^k^	88.32 ± 28.33	90.87 ± 18.44 ^l^
	GSH [nmol/mg protein]	9.54 ± 3.91	4.39 ± 2.74 ^m^	4.39 ± 2.13 ^n^	13.49 ± 8.59 ^o^
	GST [nmol/min/mg protein]	3.70 ± 1.12	2.29 ± 1.45	3.08 ± 2.03	3.90 ± 2.05
	TEAC [nmol/mg protein]	49.25 ± 23.16	35.07 ± 3.57	42.25 ± 16.73	37.23 ± 6.46
	TBARS [nmol MDA/mg protein]	0.66 ± 0.14	0.58 ± 0.08	0.45 ± 0.21	0.51 ± 0.15
	NO [nmol/mg protein]	0.49 ± 0.26	0.37 ± 0.11	0.47 ± 0.17	0.48 ± 0.10

## Data Availability

The data presented in this study are available in the article.

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
