# Peer review of "Oxidative Stress in Long-Term Exposure to Multi-Walled Carbon Nanotubes in Male Rats"

_antioxidants, 2023, doi:10.3390/antiox12020464_

Round 1

Reviewer 1 Report

The manuscript is very well written and a bit long, it contains a lot of information, close to a review at some points, which can be very useful but mask the real data, which are not many.

The work is well designed and executed, and the parameters chosen to analyze and to evaluate the level of oxidation in relevant organs for the detoxification of the organism as well.

It is a good and interesting article, but with two major flaws, in my opinion, that I think should be highlighted in the discussion and should be reflected in the final analysis more clearly, so as to allow the reader to know exactly the importance of this work.

The first one of these is the low number of animals per group, if I understood correctly there are only 6 mice per group, which, with the high deviations that exist in the numbers of some of the parameters analyzed, I think it is risky to draw conclusions.

In other words, when the data are significant, as it happens in various cases with the

18-month analysis, there is no doubt, but when they are not significant and the deviations, or standard error, I don't know what it is, are so big, it is risky to conclude that there is no effect.

I believe that the data, from so few animals, must be handled with care. I am perfectly aware of the difficulty of these experiments in the long term, so I believe that some comments should be softened and I do not believe that it is necessary to repeat them or increase the now.

The other point that seems difficult to accept and should be clearly outlined  and taken into account in the discussion, is the fact that only male animals are analyzed. I am also aware of the complexity of including female animals in these analyses, but this should be done and I believe that it deserves a clear explanation of why they have been excluded. I believe that this fact should be discussed and put into perspective and give a plausible explanation or an acceptable justification to the exclusion of adult female animals in the work, to see whether or not there was a change compared to the males.

Reviewer 2 Report

The paper of Florek et al evaluate selected biochemical parameters reflecting the intensity of oxidative stress in serum, liver and kidney tissue among the young male rats after 12 and 18 months of administration with a single injection to the knee joint with a solution of multi-walled carbon nanotubes (MWCNT).

The study present some problems listed below. Moreover, in whole manuscript there are many mistakes of acronyms. The acronyms mast be checked in manuscript.

Title

The experimental design regards male rats only,  the words “Animal Model” should be change with “male rats”.

Abstract

The nanodimension of the MWCNT should be insert.

The aim of work lacking, please clarified and insert aim in the abstract. Insert the key results.

Introduction

The introduction is very confused and long, contains repetitive text and some description (liver, oxidative stress, toxicity of nanoparticles) are too generic and didactic.

The aim of the study is lacking. 

The authors have to synthesize the whole introduction, should be focalized on MWCNT toxicity and define a clear aim.

Moreover, when the authors describe literature studies of MWCNT toxicity the dimension of nanoparticles should be reported and the data gap should be identified.

Animals and experimental treatments

A figure of experimental design could be insert to better explain the animal study.

Some question should be clarified and the answers should  be inserted in the text:

Why the authors use male rats just weaned (4 weeks old)?

Why the authors used male rats only?

Why the water was sterilized before being given to the male rats?

Explanation of number of animals for group. Why the authors used 6 rats for group? How the groups size was calculated?

The explanation/rational of concentrations selection should be insert in the text. How the solutions of MWCTN were prepared?

Why the authors administration of MWCTN to the knee joint? 

Which liver lobes were used for hystological analysis and which liver lobs were used  to measure the enzymatic activity? How kidneys were divided for endpoints analysis?

During the experimental procedures, were rats monitored for general health conditions/status? Were body weight and food consumption of rats measured? Were the organs weight measured? In the result section the general toxicity data (body weight, food consumption, organs absolute e relative weight) must be insert.

Why the CAT activity were evaluated in liver and kidney only?

Histological analyses 

How the scoring of histologic lesions were determinate?

The results of histological analysis could be showed with a table or figure

Concentrations of oxidative stress markers and biochemical parameters 

The results section should be summarize and improved.

In the text should be reported the significant results only. The results not significant are present in table and figures only. Moreover, in the text should be report 

-significant results of treatment groups vs control group at similar time

-significant results of treatment group (0, 0.03, 0.25 and 0.5) at 12 vs 18 months.

The statistical difference between treatment group should be reported in table and figures only.

Lines 416-420 should be deled. The indication of table and figures are reported in the text.

In my opinion the table 1 is difficult to read. The authors could be improved the table 1, e.g. reporting 4 concentrations (0, 0.03, 0,25 and 0.5), for 2 times (12 and 18 months) on column and the endpoints of oxidative stress on lines.

Lines 484-486 should be deleted. The results describe serum TP levels.

Lines 492-496. Where is the statistically significance of 0.5 vs control group at 18 month in figure 2?

Lines 511-513. Where is the statistical significance in figure 2?

Figure 3 “a - statistically significant” must be moved above the histogram of kidneys control

Figura 4. "a-statistically significant difference for group receiving MWCNT in concentration of 0.5 mg/mL versus 0.25 600 mg/mL in the kidneys (p=0,025661)". This sentence is in contrast with sentence of lines 616-619. The authors must be clarified this point.

Discussion-Conclusion- Summary

The Discussion-Conclusion- Summary is very long and complex, the authors could synthetize it highlighting important concepts only, and the conclusion must be in line with the study results. A lot results are repeated e.g. lines 861-862, lines 876-882 etc. The authors should deleted all result repetition and reported the key points only. Moreover, the discussion should be focalized on the few results than showed a statistical significant between control groups and treatments groups. At 0.03 there aren't statistical significant with control groups. In the serum, at 12 months, the results showed a dose-response of TP levels, like at NO levels in kidneys at 18 months, significant at 0.5. This points should be discussed.

The experimental design involves the administration by injection in knee joint, why? This point should be introduced and discussed. Young male rats were used, also this point should be introduced and discussed.

Round 2
